# A hybrid stochastic-deterministic approach to explore multiple infection and evolution in HIV

**Jesse Kreger**[1,2]*, **Natalia L. Komarova**[2], **Dominik Wodarz**[2,3]

**1** Department of Quantitative and Computational Biology, University of Southern California, Los Angeles, California, United States of America, **2** Department of Mathematics, University of California Irvine, Irvine, California, United States of America, **3** Department of Population Health and Disease Prevention Program in Public Health Susan and Henry Samueli College of Health Sciences, University of California, Irvine, California, United States of America

\* jessekre@usc.edu

**Data Availability Statement:** All relevant data are within the manuscript and its Supporting information files.

**Funding:** This work was supported by National Science Foundation (https://www.nsf.gov/) grant

## Abstract

To study viral evolutionary processes within patients, mathematical models have been instrumental. Yet, the need for stochastic simulations of minority mutant dynamics can pose computational challenges, especially in heterogeneous systems where very large and very small sub-populations coexist. Here, we describe a hybrid stochastic-deterministic algorithm to simulate mutant evolution in large viral populations, such as acute HIV-1 infection, and further include the multiple infection of cells. We demonstrate that the hybrid method can approximate the fully stochastic dynamics with sufficient accuracy at a fraction of the computational time, and quantify evolutionary end points that cannot be expressed by deterministic models, such as the mutant distribution or the probability of mutant existence at a given infected cell population size. We apply this method to study the role of multiple infection and intracellular interactions among different virus strains (such as complementation and interference) for mutant evolution. Multiple infection is predicted to increase the number of mutants at a given infected cell population size, due to a larger number of infection events. We further find that viral complementation can significantly enhance the spread of disadvantageous mutants, but only in select circumstances: it requires the occurrence of direct cell-to-cell transmission through virological synapses, as well as a substantial fitness disadvantage of the mutant, most likely corresponding to defective virus particles. This, however, likely has strong biological consequences because defective viruses can carry genetic diversity that can be incorporated into functional virus genomes via recombination. Through this mechanism, synaptic transmission in HIV might promote virus evolvability.

## Author summary

The evolution of human immunodeficiency virus within patients is an important part of the disease process. In particular, the presence of mutants that are resistant against anti-viral drugs can result in challenges to the long-term control of the infection. To study

number DMS 1662146/1662096 which was awarded to NLK and DW. We also acknowledge support from the National Science Foundation Simons Center for Multiscale Cell Fate Research. The funders had no role in study design, data collection and analysis, decision to publish, or preparation of the manuscript.

**Competing interests:** The authors have declared that no competing interests exist.

disease progression, computer simulations have been useful. However, in some cases these simulations can be difficult because of the complexity of the model. Here, we use a computational complexity reducing algorithm to simulate mutant dynamics in large populations, which can approximate the full model at a fraction of the time. The use of this algorithm allows us to study different transmission methods, viral processes that occur between virus strains within individual cells, and important quantities such as the mutant distribution or the probability of mutant existence at a given infected cell population size. We find that the direct synaptic cell-to-cell transmission of the virus through virological synapses can have strong biological consequences because it can promote potentially defective viruses that carry genetic diversity which can be incorporated into functional virus genomes during infection. Through this process, synaptic transmission in human immunodeficiency virus might promote virus evolvability.

## 1 Introduction

The evolution of HIV-1 within patients is an important determinant of the disease process and of treatment outcomes [1–3]. Evolutionary changes in the virus population over time are thought to contribute to the progression of the infection from the asymptomatic phase towards AIDS [1], involving the evolution of immune escape as well as evolution towards faster replication, increased cytopathicity, and broader cell tropism [1]. The emergence of mutants that are resistant against anti-viral drugs can result in challenges to the long-term control of the infection. While viral evolution is important throughout the course of the disease, extensive virus replication towards relatively high viral loads during the acute phase of the infection presents ample opportunity for the generation of viral mutants that might influence post-acute setpoint virus load, the subsequent disease course, and the response to treatment [4].

Mathematical models have played a key role in defining the principles of within-host dynamics and evolution of HIV [5–13]. During the acute phase of the infection, however, the number of virus-infected cells can reach very large numbers [14, 15], while some sub-populations of importance may be very small, which presents computational problems. Mutant evolution can be driven by stochastic effects, because mutant viruses initially exist at low population sizes, even though the wild-type population can be very large. Stochastic simulations of the viral evolutionary dynamics thus become computationally costly if the overall viral population size is large. To get around this, models can assume unrealistically low population sizes of infected cells, together with unrealistically large mutation rates, in the hope that the effects observed in such models scale up to more realistic population sizes and lower mutation rates. The accuracy of such explorations, however, is unclear. Alternatively, deterministic models in the form of ordinary differential equations (ODEs) can be used to approximate the average number of mutants over time as the virus population grows. The disadvantage of this approach is that other important evolutionary measures, such as the number of mutants at a given infected cell population size or the time of mutant generation, are not clearly defined in ODEs. Furthermore, the distribution of mutants at a given time or at a given infected cell population size cannot be determined with ODEs.

An interesting aspect that can influence the viral evolutionary dynamics, especially at large population sizes [16], is the multiple infection of cells [17–20]. Multiple infection has been documented to occur with HIV both in vitro [21, 22] and in vivo from human tissue samples [23]. Multiple infection is especially promoted by direct cell-to-cell transmission of the virus through virological synapses [24–27]. In this process multiple viruses are likely transferred

from the source cell to the target cell (this is in contrast to free virus transmission, where offspring virus is released into the extracellular environment, typically leading to the infection of the target cell with a single virus). The process of synaptic transmission has been well documented experimentally [24, 25], and experiments with humanized mice indicate the importance of this process in vivo [23]. As the virus grows to high levels, minority populations of multiply infected cells, which can be governed by stochastic effects, coexist with a much larger population of singly infected cells, which is again a computationally challenging situation. Interesting evolutionary dynamics can occur as a result of multiply infected cells, especially if different virus strains are present in the same cell. A disadvantageous mutant can gain fitness through complementation [28], and an advantageous mutant might experience fitness reduction due to interference by the wild-type virus [29]. Recombination can be another evolutionary consequence of multiple infection [16, 17, 30, 31].

In this paper, we present a computational study of the evolutionary dynamics of an in vivo virus infection model that contains both small and large populations simultaneously, where stochastic fluctuations of minority mutant populations can determine the end result of the system and the evolutionary potential of an infection. In classical fully stochastic algorithms like Gillespie's method, the average time step decreases as the population size increases [32], and therefore in order to calculate different important evolutionary measures, we turn to a hybrid stochastic-deterministic algorithm that is based on our previous work, applied to a different system in the field of mathematical oncology [33]. This algorithm was specifically developed to handle large population dynamic models where very small (e.g. rare mutants) and very large populations co-exist and interact. This algorithm has the advantage of intuitive transparency and computational efficiency.

We take advantage of the power of this algorithm to explore questions about evolutionary dynamics of HIV, especially in the context of multiple viral infection, and the different infection pathways (free-virus vs synaptic transmission). This includes an analysis of how intracellular interactions among viruses, such as complementation and interference, can influence evolutionary trajectories. In this context, special emphasis is placed on the direct cell-to-cell transmission of HIV through virological synapses [24–27], because it has been shown that synaptic transmission not only promotes multiple infection, but can promote the repeated co-transmission of genetically distinct virus strains from one cell to the next. This in turn can enhance the potential of complementation and interference to impact mutant spread. In this paper, we do not focus on recombination processes, which were analyzed in a previous paper [31].

This paper makes two contributions: (i) We describe a stochastic-deterministic hybrid method, which allows us to simulate the evolutionary dynamics of viruses at large population sizes (but in the presence of small subpopulations of evolutionary importance), including the possibility of multiple infection of cells. (ii) We apply this methodology to investigate the effect of multiple infection on mutant evolution during acute HIV infection. The paper starts by describing the basic mathematical model under consideration. This is followed by a description of the stochastic-deterministic hybrid methodology and a comparison of simulation results to both fully stochastic simulations and ODEs. Finally, we apply the hybrid methodology to study viral evolutionary dynamics during acute HIV infection, in the presence and absence of multiple infection, focusing on the role of viral complementation and interference. This work has relevance beyond HIV, because multiple infection and intracellular interactions (such as complementation and interference) can occur in other viruses, such as bacteriophages [29, 34]. Therefore, beyond parameter combinations that are relevant to HIV, we also explore wider parameter sets for broader relevance.

**Table 1. Description of model parameters and units (if applicable).**

| Notation | Description | Units (if applicable) |
|---|---|---|
| $\lambda$ | production rate of uninfected cells | days$^{-1}$ |
| $\beta$ | rate of free virus transmission | days$^{-1}$ |
| $\gamma$ | rate of synaptic cell-to-cell transmission | days$^{-1}$ |
| $d$ | death rate of uninfected cells | days$^{-1}$ |
| $a$ | death rate of infected cells | days$^{-1}$ |
| $x_0(t)$ | number of uninfected cells at time $t$ | cell numbers |
| $x_i(t)$ | number of cells infected with $i$ copies of the virus at time $t$ | cell numbers |
| $Z(t)$ | sum of all infected populations at time $t$, $Z(t) = \sum_{i=1}^{N} x_i(t)$ | cell numbers |
| $Z_i(t)$ | sum of fraction of subpopulations infected with $i^{\text{th}}$ strain | cell numbers |
| $\mathcal{M}$ | hybrid algorithm size threshold | cell numbers |
| $N$ | maximum infection multiplicity | virus numbers |
| $S$ | number of viruses transferred per synapse | virus numbers |
| $\mu$ | mutation rate | NA |
| $F_i$ | fitness of the $i^{\text{th}}$ strain | NA |

## 2 Methods

### 2.1 Mathematical model description: One viral strain

We begin with a deterministic model for HIV-1 infection, which includes both free virus transmission and synaptic cell-to-cell transmission [7]. We assume that cells are sufficiently well-mixed, such that relative spatial locations of cells do not play a significant role in the dynamics. To include the possibility of multiple infection, we let $x_i(t)$ represent the number of cells infected with $i$ copies of the virus at time $t$, where $i$ ranges from 0 (uninfected cells) to $N$ (cells infected with $N$ viruses). Descriptions of the model parameters can be found in Table 1. With only one viral strain, the ODE model (in its simplest formulation) is

$$\dot{x}_0 = \lambda - \beta Z x_0 - \gamma Z x_0 - dx_0, \tag{1}$$

$$\dot{x}_i = \beta Z(x_{i-1} - x_i) - \gamma Z x_i - ax_i, \ \text{if} \ 0 < i < S, \tag{2}$$

$$\dot{x}_i = \beta Z(x_{i-1} - x_i) + \gamma Z(x_{i-S} - x_i) - ax_i, \ \text{if} \ S \le i \le N - S, \tag{3}$$

$$\dot{x}_i = \beta Z(x_{i-1} - x_i) + \gamma Z x_{i-S} - ax_i, \ \text{if} \ N - S < i < N, \tag{4}$$

$$\dot{x}_N = \beta Z x_{N-1} + \gamma Z x_{N-S} - ax_N, \tag{5}$$

where the number of infected cells is defined as

$$Z(t) = \sum_{i=1}^{N} x_i(t). \tag{6}$$

We assume that $N$, the maximum multiplicity of infection, is large enough to not result in a significant amount of cells near the end of the infection cascade. In the above equations, cell free virus transmission happens at rate $\beta$, and synaptic transmission (whereby $S$ viruses are transmitted from a donor cell to a target cell) at rate $\gamma$. Terms containing $\beta$ represents the rate at which a cell of type $x_i$ (with $0 \le i < N$) can become (super)-infected by means of free-virus

transmission, at a rate proportional to $Z$, which comprises all subpopulations infected with 1, 2, ... copies of virus. It is assumed that in quasi-steady state the number of free viruses is proportional to the total population size of infected cells, $Z$ (see Section 1.2 of S1 Text for details). As a result, a cell of type $x_i$ becomes a cell of type $x_{i+1}$. Mathematically, the process of synaptic transmission is similar, except that free virus transmission involves the entry of one virus into the target cells, while multiple viruses (e.g. $S$ viruses) can enter the target cell simultaneously during synaptic transmission. Therefore, synaptic infection (terms multiplying $\gamma$) result in a cell of type $x_i$ becoming a cell of type $x_{i+S}$; see the next section for for a more general model.

This model has a virus free steady state,

$$x_0 = \frac{\lambda}{d}, \quad Z = 0, \tag{7}$$

and an infection steady state,

$$x_0 = \frac{a}{\beta + \gamma}, \quad Z = \frac{\lambda}{a} - \frac{d}{\beta + \gamma}. \tag{8}$$

The stability of these steady states depends on the basic reproductive ratio, $R_0 = \frac{\lambda(\beta+\gamma)}{ad}$. If $R_0 < 1$, the virus free steady state is stable and if $R_0 > 1$ then the infection steady state is stable. Therefore, when considering the total virus population, the properties of this model are identical to those in standard virus dynamics models [5, 7].

## 2.2 Mathematical model with multiple viral strains

This model can be adapted to describe competition among different virus strains, and mutational processes that give rise to mutant viral strains, thus allowing us to study the evolutionary dynamics of the virus.

For neutral mutants, the rate of virus transmission from a multiply infected cells is proportional to the fraction of the virus strain in the infected cell. For advantageous or disadvantageous mutants, this also applies. Fitness differences are modeled by modifying the probability of the virus strain that has been chosen for infection to successfully enter the new target cell (note that the basic formulation (1–5) assumes that viruses are 100% successful in infecting the target cell). For example, a disadvantageous mutant is assumed to have an increased probability that successful infection fails. Hence, fitness differences are expressed at the level of entry into the new target cells. Mutations are assumed to occur during the infection process, corresponding to mutations that occur during reverse transcription in HIV infection. We refer to "mutants" as virus strains with a specific characteristic, such as a drug-resistant virus strain, an immune escape strain, or another specific phenotype. We refer to the virus population that does not share this characteristic as the non-mutant or wild-type population, even though RNA virus populations tend to exist as a quasi-species, due to reduced replication fidelity [35]. Next we derive the ODEs describing virus dynamics in the presence of multiple strains.

Assume that we have two strains, the wild-type and mutant. In order to model synaptic transmission with multiple strains and fitness considerations, we start by considering an infecting cell that contains $n$ wild-type viruses and $m$ mutant viruses, where $0 < n + m \leq N$. We denote the fitness of the wild-type as $F_1$ and fitness of mutant as $F_2$, where these parameters have the meaning of the probability of successful infection, i.e. $0 \leq F_1, F_2 \leq 1$. Here $F_2$ could be smaller (disadvantageous mutant), equal (neutral mutant), or larger (advantageous mutant) than $F_1$. Let us denote the fraction of wild-type and mutant viruses as

$$v = \frac{n}{n + m}, \quad \psi = \frac{m}{n + m},$$

respectively. Synaptic transmission is modeled as follows. We fix the number of viruses that are picked up for a synaptic transmission event, $S = 3$ (free virus transmission is similar, only with $S = 1$) [36]. Then, the following procedure is repeated $S$ times: a virus is selected from the infecting cell with the probability equal to its abundance in the cell (that is, wild-type viruses are picked with probability $v$ and mutants with probability $\psi$). Each virus that is picked will proceed to infect the target cell successfully with the probability given by its fitness (that is, $F_1$ for the wild-type and $F_2$ for the mutant). Each "pick" can result in three possibilities:

1. A wild-type virus will go on to be successful in infecting the target cell; this happens with probability $p_1 = vF_1$. We denote that by $*$ below.

2. A mutant virus will go on to be successful in infecting the target cell; this happens with probability $p_2 = \psi F_2$. We denote that by $X$ below.

3. An unsuccessful infection event, which happens with probability $p_3 = v(1 - F_1) + \psi(1 - F_2)$. We denote that by $0$ below.

Therefore, under $S = 3$, a single synaptic transmission event can result in ten different infection events. Four of them $\{* * *, **X, *XX, XXX\}$ result in an infection of the target cell with all $S = 3$ viruses (and these are the only events if $F_1 = F_2 = 1$). The other six events $\{**0, *X0, XX0, *00, X00, 000\}$ result in an infection event with fewer than $S$ viruses. The probabilities of these events can be calculated by using multinomial distributions. In particular, given that the infecting cell is characterized by $(n, m)$, the probability of an event where $\hat{s}_1$ wild-type viruses and $\hat{s}_2$ mutant viruses go on to successfully infect the target cell is given by

$$P_{n,m}(\hat{s}_1, \hat{s}_2) = \frac{S!}{\hat{s}_1! \hat{s}_2! (S - \hat{s}_1 - \hat{s}_2)!} p_1^{\hat{s}_1} p_2^{\hat{s}_2} p_3^{S - \hat{s}_1 - \hat{s}_2}. \tag{9}$$

Note the following special cases. If the target cell has $n = 0$ (that is, it is only infected by the mutant), then $\psi = 1$ and $p_2 = F_2$. The only event with $S$ successful infections is $XXX$ and it happens with probability $F_2^S$. On the other hand, if $m = 0$, we have event $* * *$ with probability $F_1^S$. In other words, fitness properties of viruses are not erased if they are in cells that are not coinfected with both virus strains.

Next, we include the process of mutations. We assume that a virus can mutate upon entering the target cell, such that the process of mutation does not affect the success of infection. As there are only two strains, denote the probability that a wild-type virus mutates by $\mu$ and the probability that a mutant back-mutates to revert to a wild-type also by $\mu$. Let us suppose that a synaptic transmission event involves $\hat{s}_1$ wild-type and $\hat{s}_2$ mutant viruses, and consider the probability that upon entering the cell, we have $\hat{i}$ wild-type and $\hat{j}$ mutant viruses, where the change is due to mutations. We denote this probability as $Q_{\hat{s}_1; \hat{i}, \hat{j}}$ (note that $\hat{s}_1 + \hat{s}_2 = \hat{i} + \hat{j}$). Suppose $\hat{a}$ out of $\hat{s}_1$ wild-type viruses mutate and $\hat{b}$ out of $\hat{s}_2$ viruses back-mutate. Then the number of (wild-type, mutant) viruses is $(\hat{s}_1 - \hat{a} + \hat{b}, \hat{s}_2 - \hat{b} + \hat{a}) = (\hat{i}, \hat{j})$. Setting $\hat{a} = \hat{s}_1 - \hat{i} + \hat{b}$, we obtain

$$Q_{\hat{s}_1; \hat{i}, \hat{j}} = \sum_{\hat{b}=0}^{\hat{s}_2} \frac{\hat{s}_1!}{\hat{a}!(\hat{s}_1 - \hat{a})!} \mu^{\hat{a}} (1 - \mu)^{\hat{s}_1 - \hat{a}} \frac{\hat{s}_2!}{\hat{b}!(\hat{s}_2 - \hat{b})!} \mu^{\hat{b}} (1 - \mu)^{\hat{s}_2 - \hat{b}}. \tag{10}$$

For the general case when any number of viruses up to general $S$ can be transmitted successfully by synaptic transmission, we have that the full model with two virus strains is

$$\dot{x}_{0,0} = \lambda - \beta x_{0,0}(Z_1 + Z_2) - \gamma x_{0,0}\left[\sum_{\hat{i}+\hat{j}\leq S}\sum_{0<n+m\leq N}\sum_{\hat{s}_1=0}^{\hat{i}+\hat{j}} P_{n,m}(\hat{s}_1, \hat{i}+\hat{j}-\hat{s}_1)x_{n,m}\right] - dx_{0,0}, \quad (11)$$

$$
\begin{aligned}
\dot{x}_{i,j} &= \beta\left[((1-\mu)Z_1 + \mu Z_2)x_{i-1,j} + (\mu Z_1 + (1-\mu)Z_2)x_{i,j-1} - (Z_1+Z_2)x_{i,j}\right] \\
&+ \gamma\left[\sum_{\hat{i}+\hat{j}\leq S}\sum_{0<n+m\leq N}\sum_{\hat{s}_1=0}^{\hat{i}+\hat{j}} P_{n,m}(\hat{s}_1, \hat{i}+\hat{j}-\hat{s}_1)Q_{\hat{s}_1;\hat{i}\hat{j}}x_{n,m}(x_{i-\hat{i},j-\hat{j}} - x_{i,j})\right] - ax_{i,j},
\end{aligned}
\quad (12)
$$

where $Z_1 = F_1\sum_{0<i+j\leq N}\frac{i}{i+j}$ and $Z_2 = F_2\sum_{0<i+j\leq N}\frac{j}{i+j}$, and with the appropriate adjustments that any population with a negative index is 0 and cells cannot be infected with more than $N$ total copies of virus. Note that in the case of only free virus transmission ($\gamma = 0$), the fitness parameters can be interpreted as factors that modulate the rate of infection $\beta$. A system with more virus strains can easily be created as a generalization of this.

The number of equations per model where mutation can happen at $k$ independent locations is $2^{-k}(N+1)\binom{N+2^k}{2^k-1}$. To see this, we note that there are $2^k$ virus strains. The number of ways to distribute $j$ viral copies into the $2^k$ strains is $\binom{j+2^k-1}{2^k-1}$. Since we allow $j \in 0, \ldots, N$, we have $\sum_{j=0}^{N}\binom{j+2^k-1}{2^k-1} = 2^{-k}(N+1)\binom{N+2^k}{2^k-1} = \binom{N+2^k}{2^k}$.

If we let $x_0$ denote the number of uninfected cells and $Z$ denote the sum of all infected cell subpopulations, we have that this generalized model again has a virus free steady state, Eq (7), and an infection steady state, which instead of Eq (8) is now given by

$$x_0 = \frac{a}{\beta F + \gamma(1-(1-F)^S)}, \quad (13)$$

$$Z = \frac{\lambda}{a} - \frac{d}{\beta F + \gamma(1-(1-F)^S)}. \quad (14)$$

The stability of these steady states depends on the basic reproductive ratio, $R_0 = \frac{\lambda(\beta F + \gamma(1-(1-F)^S))}{ad}$. Again we have that if $R_0 < 1$ the virus free steady state is stable and if $R_0 > 1$ then the infection steady state is stable.

In computer simulations, we will concentrate on parameters that are relevant for acute HIV infection, characterized by a basic reproductive ratio $R_0 = 8$. The assumed model parameters are based on the literature and explained in S1 Text Section 1.1. Since the model is applicable to viruses other than HIV, we also vary parameters more broadly to investigate dynamics for lower values of $R_0$, where we expect to see larger effects of stochasticity.

## 2.3 Hybrid algorithm

Here, we describe a stochastic-deterministic hybrid algorithm that simulates the dynamics of small mutant populations and small populations of multiply infected cells stochastically, while describing the majority populations deterministically. This allows us to run computationally efficient simulations of viral evolutionary processes at large population sizes, without losing the effects arising from the stochastic dynamics of minority subpopulations.

This methodology is based on our previous work in the context of tumor cell evolution [33], which in turn is related to work in the field of chemical kinetics [37–39]. Recently, and especially in the field of physical chemistry, many innovative computational algorithms have been developed to simulate stochastic systems, which can result in significant speed improvements and other advantages compared to the basic Gillespie algorithm [32]. Such methods include the next reaction method and tau-leaping methods (or adaptive tau-leaping methods, which features an adaptive step size) [40], which can potentially provide a large computational advantage over the Gillespie method by taking much larger steps in time while still capturing important stochastic effects by assessing how many times each stochastic reaction "fires" in the relevant time interval. However, the existence of both small and large populations of importance (and/or when the reaction propensities are highly dynamic and change quickly) generally implies that methods such as tau-leaping will be inefficient [41]. Furthermore, when different populations and reaction propensities differ over several orders of magnitude, measuring how many times a reaction "fires" in a given interval is somewhat counterintuitive. To this end, there has also been a focus on the development of novel hybrid stochastic-deterministic approaches, including many different multi-scale methods that are designed to simulate systems that contain different time, size, and spatial scales [41–49].

Much important work has also been done on the mathematical properties and analysis of stochastic multi-scale models, including in [43, 50–52]. In particular, [50] provides detailed mathematical justification and motivation for the use of stochastic continuous-time Markov chains in simulating chemical networks, as well as analytic approaches for model approximation/reduction techniques for complex systems. Ref. [43] provides a theoretical foundation for a class of stochastic models, specifically where some populations and/or reaction propensities vary in size over several orders of magnitude. One such example is the use of a pre-existing model of a cell's viral infection to demonstrate the power of model dimensionality reduction techniques for mathematically analyzing complex networks. Specifically, a combination of averaging and law of large number arguments is used on the viral dynamics model (which overall has three populations and six stochastic reactions) to 1) show that the "slow" component of the model can be approximated by a deterministic equation, 2) characterize the asymptotic distribution of the "fast" components, and 3) derive important quantities such as the probability a single virus successfully infects a cell and the expected time until the establishment of an infection. These such arguments explicitly provide theoretical justification for the identification of different scales in multi-scale analysis of stochastic models.

While similar model reduction approaches and algorithms are often used in the field of physical chemistry, they are less common in the fields of population dynamics and evolution, as they can rely on theoretical physical concepts such as Langevin's equation. In this paper, we choose to implement the hybrid methodology described in [33], which partitions the system into a large and small component through the use of an explicit size threshold. This is because our evolutionary system under consideration contains a large overall population size and number of reactions, random and rare mutation events, and the simultaneous existence of both large and small populations of importance. The use of the hybrid method allows us to run computer simulations of the system at a reasonable computational expense.

Our hybrid algorithm is based on the idea that if a cell population is sufficiently large, an ODE representation can provide a good approximation of most stochastic trajectories of the population. We can write the ODE system as a single vector equation $d\mathbf{V}/dt = \mathbf{F}(\mathbf{V})$, where $\mathbf{V}$ is a vector that contains all the cell subpopulations. Let $\mathcal{M}$ be a given population size threshold, that applies to all subpopulations. We classify each cell population $x_i$ as small at time $t$ if $x_i(t) < \mathcal{M}$, or large otherwise. The classification of small versus large populations is re-checked at every step in the algorithm. We simulate the small populations stochastically using

the Gillespie algorithm and use the ODEs for the large populations. Further details of the hybrid method are given in S1 Text Section 2.

**2.3.1 Implementation.**    The size threshold $\mathcal{M}$ is a very important parameter in the hybrid algorithm. If $\mathcal{M} = 0$, then at each time point every non-zero population is classified as large and the hybrid algorithm is identical to the deterministic solution of the ODEs. If $\mathcal{M}$ is very large, that is larger than all populations for the duration of the time-span of interest, then the hybrid algorithm is the same as the completely stochastic Gillespie simulation of the model and can be extremely computationally inefficient. For intermediate $\mathcal{M} > 0$, the hybrid algorithm is computationally efficient and the averages over many hybrid simulations go from approximating the deterministic predictions to converging to the stochastic averages as $\mathcal{M}$ increases. Therefore, in order to efficiently approximate the completely stochastic implementation of the model, we need to choose an intermediate $\mathcal{M}$ such that the results are close to the fully stochastic implementation.

We can achieve this by comparing the hybrid averages over many simulations to completely stochastic averages over many simulations for simplified models, such as assuming a constant large number of uninfected cells or using parameter values that result in smaller and more computationally manageable population sizes. For these models, completely stochastic simulations can be carried out and allow us to determine what size threshold $\mathcal{M}$ is reasonable for the related models. Specifically, since the averages over many hybrid simulations start from the deterministic prediction ($\mathcal{M} = 0$) and converge to the completely stochastic average, similarly to [33] we i) set some difference threshold $\varepsilon > 0$, ii) test multiple size thresholds $\mathcal{M}$, and iii) choose the smallest $\mathcal{M}$ such that the hybrid average is within $\varepsilon$ of the completely stochastic average for the relevant mutant strains and/or subpopulations.

Table 2 contains approximate computer simulation run times for the completely deterministic ODE system, the hybrid method, and the completely stochastic Gillespie algorithm (for comparison with the tau-leaping method, see Section 2.4 of S1 Text). Each system is run for the single mutation, double mutation, and triple mutation models. All simulations include only free virus transmission with limited multiple infection ($N = 3$), represent established infections only (we ignore stochastic simulations in which the infection dies out), and are stopped once the infected cell population reaches $10^8$ cells. The times for the ODE and hybrid simulations also depend on the ODE solution method and the step size, $h$ (here $h = 10^{-5}$ with

**Table 2. Approximate average run times for a single simulation for the completely deterministic ODE system (Euler method with step-size $h = 10^{-5}$), the hybrid method with different threshold values ($\mathcal{M}$), and the completely stochastic Gillespie algorithm (rows). Each system is run for the single mutation, double mutation, and triple mutation models (columns). In each system we assume all strains are neutral ($F_i = 1$ for all $i$). The other parameters are $N = 3$, $\mu = 3 \times 10^{-5}$, $\lambda = 1.59 \times 10^7$, $\beta = 3.60 \times 10^{-9}$, $\gamma = 0$, $d = 0.016$, and $a = 0.45$.**

| Model | single mutation $k = 1$ 2 strains, 10 equations | double mutation $k = 2$ 4 strains, 35 equations | triple mutation $k = 3$ 8 strains, 165 equations |
|---|---|---|---|
| Full ODEs | < 1 second | 4 seconds | 12 minutes |
| Hybrid, $\mathcal{M} = 10$ | < 1 second | 4 seconds | 13 minutes |
| Hybrid, $\mathcal{M} = 10^3$ | < 1 second | 4 seconds | 13 minutes |
| Hybrid, $\mathcal{M} = 10^5$ | < 1 second | 6 seconds | 15 minutes |
| Hybrid, $\mathcal{M} = 10^7$ | 1 minute | 7 minutes | 30 hours |
| Full Gillespie | 12 minutes | 100 minutes | 1 week |

Euler method). In general, with $k$ possible mutations, the number of strains per model is $2^k$ and the number of equations (subpopulations) per model is $\left(\frac{N+2^k}{2^k}\right)$.

Because the parameters chosen for the simulations in Table 2 correspond to $R_0 = 8$, a relatively small size threshold $\mathcal{M}$ gives a good approximation of the fully stochastic simulations. Simulations with lower $R_0$ require higher values of $\mathcal{M}$ and hence take longer to run.

**2.3.2 Choosing a size threshold $\mathcal{M}$.**   We have developed an analytical method for finding a lower bound on size threshold $\mathcal{M}$, which is based on the notion of $R_0$. This method does not depend on the number of mutations, infection multiplicity, fitness landscape, etc. The basic reproductive ratio, $R_0$, is the average number of newly infected cells generated per single infected cell at the beginning of the infection. Therefore, infections with larger $R_0$ will lead to quicker and more successful growth of the overall virus population. While in a deterministic system, infections with $R_0 > 1$ will never go extinct, in the stochastic setting, even if $R_0 > 1$, a single infected cell can die out before successfully infecting other cells. The rate at which infections stochastically go extinct is given by $\frac{1}{R_0}$ [53, 54]; in other words, infection will successfully spread with probability $\Phi^\infty = 1 - 1/R_0$. Moreover, one can show that an infection will increase until size $K$ (before possibly going extinct) with probability

$$\Phi^K = \frac{1 - \dfrac{1}{R_0}}{1 - \left(\dfrac{1}{R_0}\right)^K}. \tag{15}$$

Setting the size-threshold to a given value $\mathcal{M}$ essentially means that we assume that a population that has reached that size will no longer go extinct, because its subsequent dynamics are described by ODEs. Let $\delta > 0$ be some small difference threshold. We define the lower bound size threshold, $\hat{\mathcal{M}}$, as the smallest natural number $\mathcal{M}$ such that

$$|\Phi^\mathcal{M} - \Phi^\infty| < \delta,$$

which gives the estimate

$$\hat{\mathcal{M}} = \lceil \ln\left(1 + \frac{R_0 - 1}{\delta R_0}\right)/\ln R_0 \rceil, \tag{16}$$

where $\lceil . \rceil$ denotes the ceiling function. Note that $\hat{\mathcal{M}}$ is a lower bound, and the calculation above is based only on the dynamics of the wild type strain, without taking into account any information on the mutant parameters. Therefore, depending on the details of the model (such as the number and type of mutant strains), it is possible that a larger $\mathcal{M}$ is needed to get accurate descriptions of mutant dynamics. In general, we can always confirm that a chosen $\mathcal{M}$ is large enough using the $\varepsilon$ test described in the preceding section and in [33].

## 3 Results

### 3.1 Comparing and contrasting ODE versus stochastic / hybrid simulations

ODE (deterministic) and stochastic modeling approaches have their advantages and disadvantages. ODE modeling is very intuitive and provides excellent insights into viral dynamics, including the expected mean trajectories of wild type and mutant population sizes. Stochastic models are much harder to implement, slow to run (thus we developed our hybrid method), but they contain more information about evolutionary dynamics. In particular, stochastic modeling allows studies of distributions (such as mutant number distributions and the

distribution of generation times). Also, stochastic models can describe the number of mutants at a given population size, or the time of mutant generation, which are not clearly defined in the continuous ODEs. In particular, if we determine the number of mutants in ODE simulations once the infected cell population size in the ODE has reached a threshold $N$ (say, at time $t_N$), we are effectively determining the average number of mutants over different stochastic trajectories, which all correspond to different infected cell population sizes. This is because at time $t_N$, while the average number of infected cells reaches size $N$, for some stochastic realizations, this number at that time will be lower and for others, higher than $N$.

To underline these points, in this section we compare ODE predictions to outputs from the stochastic simulations, in the context of the evolution of neutral, advantageous, and disadvantageous mutants. Here we focus on relatively simple scenarios, considering the exponential growth phase of the virus population and only including free virus transmission; synaptic transmission and infection peak dynamics are studied in the next section. While parameter sets explored here are relevant to HIV, we also include broader parameter sets for comparison, especially those where the basic reproductive ratio is lower. In these regimes, the dynamics are governed by stochasticity to a larger extent.

**3.1.1 The average number of mutants at a given infected cell population size.** We start by determining the average number of neutral mutants once the number of infected cells has reached a threshold size in the purely stochastic process (we discard simulations in which the infection goes extinct stochastically before reaching the threshold size). We then compare this to the number of mutants predicted by the ODE at the time when the average infected cell population size is the same threshold. To be able to run fully stochastic simulations, we determine the number of mutants at a relatively low infected cell population size of $10^4$.

Fig 1A shows the results for a neutral mutant, assuming different values for the basic reproductive ratio of the virus, $R_0$. The lower the value of $R_0$, the higher the discrepancy between the

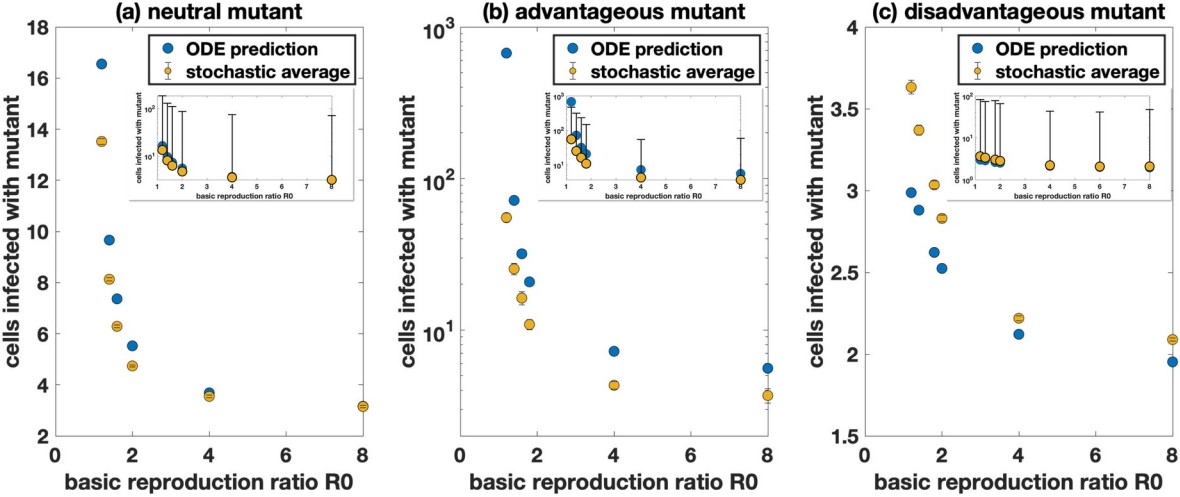

**Fig 1. Comparison of the deterministic prediction and stochastic average of the number of cells infected with the mutant with free virus transmission only.** The deterministic predictions are in blue and the stochastic hybrid simulations with $\mathcal{M} = 10^4$ (infected populations always treated completely stochastically) are in yellow. Standard error bars are included in the main panel (sometimes too small to see) and the inserts show standard deviation bars. A Neutral mutant, $F_{\text{mutant}} = 0.9$. Each yellow dot represents the average taken over at least $2 \times 10^6$ simulations. B Advantageous mutant with 10% advantage, $F_{\text{mutant}} = 0.99$. Each yellow dot represents the average taken over at least $1.1 \times 10^3$ simulations. C Disadvantageous mutant with 10% disadvantage, $F_{\text{mutant}} = 0.81$. Each yellow dot represents the average taken over at least $3.5 \times 10^6$ simulations. We have $R_0 = \frac{\lambda(\beta F + \gamma(1-(1-F)^S))}{ad}$, and the parameters are $F_{\text{wild-type}} = 0.9$, $N = 3$, $\mu = 3 \times 10^{-5}$, $\lambda = 1.59 \times 10^7$, $\beta = 4 \times 10^{-9}$, $\gamma = 0$, and $d = 0.016$. The infected cell death rate $a$ is adjusted to achieve the required $R_0$.

average of the stochastic simulations and the ODE results. For $R_0 = 8$, which is characteristic of HIV infection [55, 56], the discrepancy is minimal. The reason is that for relatively large values of $R_0$, the variation of the infected cell population size at a given time is reduced. Fig 1B and 1C show equivalent plots for advantageous and disadvantageous mutants, respectively. Again, the extent of the discrepancies increases with lower values of $R_0$. Discrepancies tend to be larger than for neutral mutants, and are apparent even for higher values of $R_0$ (e.g. $R_0 = 8$).

While ODEs cannot accurately describe the average behavior of the stochastic model, the hybrid method (with a sufficient size threshold) is able to do so, as is demonstrated in S6A Fig.

**3.1.2 The timing of mutant emergence.**   Another important measure is the time at which the first copy of a given mutant is generated, and the infected cell population size at which this mutant is generated. The closest measure in the ODE is the the time and infected cell population size at which the average number of mutants crosses unity. As shown in S5 Fig, however, significant discrepancies exist between this ODE measure and the accurate prediction of stochastic simulations, and this discrepancy increases with a larger number of mutation events required to generate this mutant (i.e. 1-hit, 2-hit. 3-hit mutants etc). The hybrid method, however, provides an accurate approximation (S6B Fig).

**3.1.3 Probability distributions of mutant numbers.**   The probability distribution of the number of mutants at a given infected cell population size, or at a given time, is a measure that has no equivalent in ODEs, yet these measures have strong biological relevance. For example, it is important to understand the likelihood that certain mutants exist at various stages during virus growth, such as virus strains resistant against one or more drugs or against one or more immune cell clones. The hybrid method provides a good approximation of the results from stochastic simulations, as shown in S3 Fig. This also applies to simulations that assume relatively low values of $R_0$ (S4 Fig), although larger size thresholds $\mathcal{M}$ are required for smaller values of $R_0$.

## 3.2 Impact of multiple infection on mutant evolution

In this section, we apply the above-described hybrid method to explore how multiple infection can affect virus evolution during an exponential growth phase and near the peak infection, with particular relevance to the acute phase of HIV infection, during which the infected cell population grows to large sizes. Multiple infection can influence viral evolution in a variety of ways. On a basic level, the ability of viruses to enter cells that are already infected increases the target cell population and allows the virus to undergo more reverse transcription events, thus increasing the effective rate at which mutations are generated. In addition, viral fitness can be altered in multiply infected cells through viral complementation or inhibition [28], which again has the potential to influence the evolutionary dynamics. In the context of HIV infection, direct cell-to-cell transmission through virological synapses (synaptic transmission) increases the complexity of these processes. Synaptic transmission typically results in the transfer of multiple viruses from the source cell to the target cell, thus increasing the level of multiple infection [24–27]. In addition, synaptic transmission can lead to the repeated co-transmission of different virus strains [21, 23] which can amplify the effect of viral complementation or inhibition. To explore these dynamics, the hybrid method is important because multiple infection becomes increasingly prevalent at large population sizes, where both mutant viruses and multiply infected cells exist as relatively small populations compared to the larger populations of wild-type viruses and singly infected cells. We will focus on basic evolutionary processes that do not involve recombination.

**3.2.1 The effect of multiple infection on the spread of neutral mutants.**   We start with the most basic scenario: the effect of multiple infection on the presence of neutral mutants

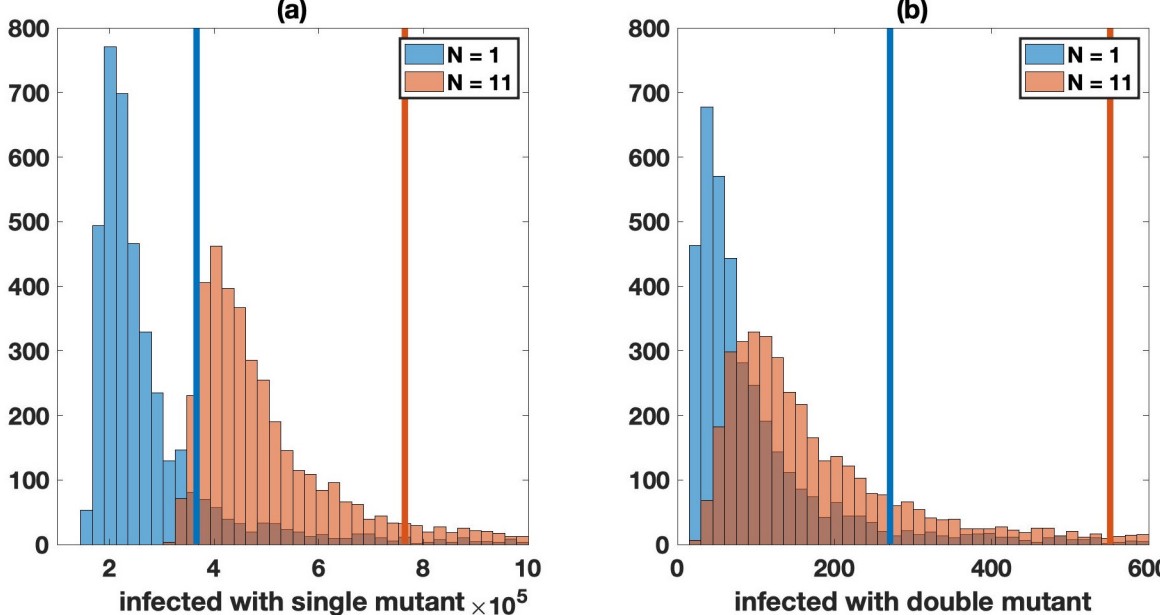

**Fig 2. Neutral mutant evolution in the absence of synaptic transmission, comparing simulations with single infection only ($N = 1$, blue) and in the presence of multiple infection ($N = 11$, red).** The mean values are shown by the vertical lines (blue for single infection only and red for multiple infection). For both panels, the Kolmogorov-Smirnov test between the two cases gives a $p$-value less than $10^{-6}$. A Number of cells infected with one of the single mutant strains. The average for single infection is approximately $3.7 \times 10^5$ and for multiple infection is approximately $7.6 \times 10^5$. B Number of cells infected with the double mutant strain. The average for single infection is approximately 271 and for multiple infection is approximately 551. Histograms represent $4 \times 10^3$ hybrid simulations with size threshold $\mathcal{M} = 50$. Simulations in which infections are not established (or in the rare case a simulation does not reach the infected size threshold) are discarded. Simulations are stopped when the infected cell population is close to peak infection ($6 \times 10^8$ cells). The other parameters are similar to Fig 1 ($F_{\text{wild-type}} = 1$, $F_{\text{mutant}} = 1$, $\mu = 3 \times 10^{-5}$, $\lambda = 1.59 \times 10^7$, $\beta = 3.60 \times 10^{-9}$, $\gamma = 0$, $a = 0.45$, $d = 0.016$, and $R_0 = 8$).

during the growth phase of the virus. For simplicity, we concentrate on free virus transmission only. Because this analysis is done with HIV in mind, we set $R_0 = 8$. S9 Fig as well as Fig 2 show histograms of cells infected with neutral single and double mutants in the presence and absence of multiple infection. S9 Fig shows that at relatively low virus loads, the average number of mutants is the same, whether multiple infection is assumed to occur or not. At larger population sizes that are close to peak virus load, however, we observe a pronounced difference, Fig 2. In these simulations, we recorded the number of mutants at $6 \times 10^8$ infected cells, as it is close to the peak and almost all stochastic simulations reached this threshold. We can see that multiple infection results in a 2-fold or larger increase in the average number of mutants, both for single-hit (Fig 2A) and double-hit mutants (Fig 2B). The reason is that larger number of infection events occur in the presence of multiple infection, thus raising the number of mutants that are generated. We further note that multiple infection not only increases the average number of mutants at high viral loads, but that it also leads to a larger variation in mutant numbers, shown by a larger standard deviation of mutant numbers in the presence of multiple infection (Fig 2).

These trends are not particular to neutral mutants because we focus on exponential, or nearly-exponential, virus growth. Similar trends are observed for advantageous or disadvantageous mutants (see S1 Text Section 4 and S10 Fig).

While computationally more costly, we also examined the prevalence of neutral triple-hit mutants, because such mutants can be important for simultaneously escaping three immune

response specificities or three drugs. We found that even near peak virus load, the probability that a triple mutant exists is relatively low (S11 Fig). In other words, such mutants are unlikely to exist even at the peak of primary HIV infection. Nevertheless, multiple infection results in an almost 2-fold increase in the probability that neutral triple mutants exist around peak infection. Such an increase in mutant generation could be important for virus persistence in the face of mounting immune responses during the acute phase of the infection.

**3.2.2 Evolutionary dynamics in more complex settings: Complementation, interference, and the role of synaptic transmission.** Multiple infection becomes especially important for viral evolutionary dynamics if different virus strains interact with each other inside the same cell. One type of such interactions is complementation, where a disadvantageous mutant gains in fitness in a coinfected cell [28]. Another example is interference, where an advantageous mutant can lose the fitness advantage when together with a wild-type virus in the same cell [29]. We will use our hybrid methodology to investigate the evolution of disadvantageous and advantageous mutants, and the effect of complementation and interference, respectively. We start by examining the dynamics assuming free virus transmission, and then compare results to simulations that assume virus spread through synaptic transmission. Synaptic transmission can be especially relevant here because it can promote the repeated co-transmission of genetically distinct virus strains. For example, if a disadvantageous mutant is repeatedly co-transmitted with a wild-type virus, and if the disadvantageous mutants benefits from complementation, then synaptic transmission can significantly enhance the spread potential of the mutant.

As before, the fitness difference is modeled at the level of the infection process. For example, for a disadvantageous mutant, there is a chance that infection of a new cell is unsuccessful. In this case, complementation means that the wild-type virus can provide a product that enhances the infectivity of the mutant. Similarly, for interference, it is assumed that the chance of infection by an advantageous mutant is reduced if the offspring mutant was generated in a coinfected cell.

**Effect of viral co-transmission on mutant spread (in the absence of mutations)**. To assess to what extent the co-transmission of different virus strains influences viral evolution, we consider computer simulations in the absence of mutant production. Instead, we start with one infected cell that contains both one wild-type and one mutant virus, and simulate the spread of the virus population until a threshold number of infected cells is reached. The purpose of excluding mutant production is to fully quantify to what extent synaptic transmission enhances the spread potential of a mutant.

Complementation: First, consider viral complementation. We study an extreme case where a mutant has zero fitness by itself, but has an infectivity identical to the wild-type virus if the mutant offspring virus is produced in a cell coinfected with a wild-type virus. In this parameter regime, the mutant virus cannot spread at all in the absence of complementation, whether spread occurs by free virus or synaptic transmission. The occurrence of complementation, however, allows virus spread due to the elevated viral fitness in coinfected cells. For free virus transmission, this effect is modest (Fig 3A). A limited amount of mutant spread can occur, but the average number of mutants at peak infection levels is still less than one, indicating that mutants largely fail to spread in this setting. In simulations with synaptic transmission, however, we observe extensive mutant spread in the presence of complementation (Fig 3B). Around peak infection, the number of cells infected with the mutant is of the order of $10^5$. This shows that synaptic transmission can play a crucial role at promoting the spread of disadvantageous mutants through complementation.

Interference: Next, consider viral interference. Assume an advantageous mutant, which has a significant fitness advantage by itself (10%), but has an infectivity identical to the wild-type

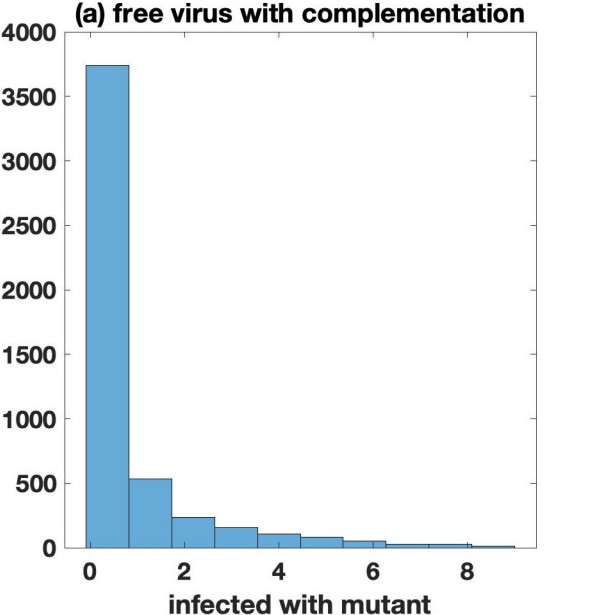
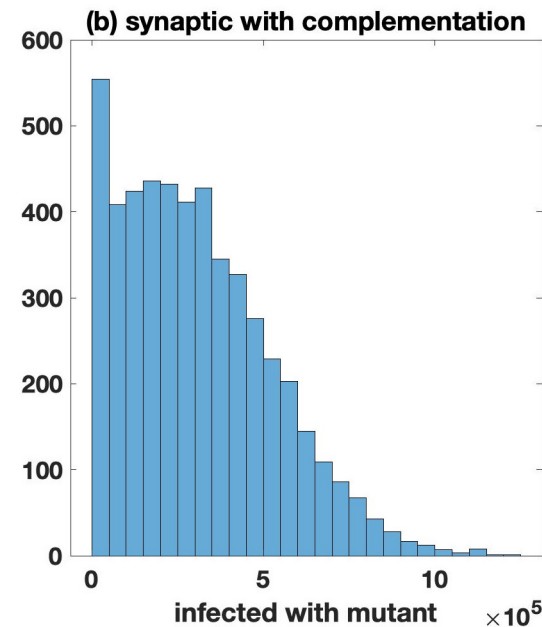

**Fig 3. Zero fitness mutants, comparing the effect of complementation for free virus and synaptic transmission.** All simulations start with a single infected cell coinfected with a single copy of both the wild-type and mutant, and mutation is turned off ($\mu = 0$). A Only free virus transmission ($\beta = 3.60 \times 10^{-9}$, $\gamma = 0$, $N = 11$) with complementation. The average number (standard deviation) of cells infected with the mutant is 0.71 (1.73). B Only synaptic transmission ($\beta = 0$, $\gamma = 3.60 \times 10^{-9}$, $N = 25$, see Section 1.3 of S1 Text for justification) with complementation. The average number (standard deviation) of cells infected with the mutant is $3.1 \times 10^5$ ($2.2 \times 10^5$). Histograms represent $5 \times 10^3$ hybrid simulations with size threshold $\mathcal{M} = 50$. Simulations in which infections are not established (or in the rare case a simulation does not reach the infected size threshold) are discarded; simulations are stopped when the infected cell population is close to peak infection ($5 \times 10^8$ cells). The fitness of the wild-type is fixed at $F_{\text{wild-type}} = 0.9$ and $F_{\text{mutant}} = 0$. The other parameters are as in Fig 1 ($\lambda = 1.59 \times 10^7$, $a = 0.45$, and $d = 0.016$).

virus if the mutant offspring is produced in a cell coinfected with the wild-type. Under free virus transmission (S13 Fig), coinfection does not play a significant role, and therefore interference only decreases the expected number of mutants by a small percentage. Interestingly, for synaptic transmission (S13 Fig), interference only plays a marginally larger role compared to the dynamics under free-virus transmission. The reason for this relatively mild effect of interference under purely synaptic transmission is rooted in an inherent reduction of fitness differences due to repeated infection events in synaptic transmission. We elaborate on this later on in the context of dynamics with mutations.

**Evolutionary dynamics in the presence of mutant production**. Here, we repeat this analysis assuming that mutant production occurs. The mutant dynamics are now influenced by two factors: (i) as before, mutant viral replication and mutant fitness influence spread; (ii) mutation processes generate mutant viruses from wild-type, which also contributes to the increase of mutant numbers. We consider both viral complementation and inhibition.

Complementation: We first focus on a mutant that has zero fitness if it is by itself in a cell. If mutant numbers are measured at relatively low virus loads (Fig 4A and 4B), complementation makes no difference for simulations that assume free virus transmission only (Fig 4A). For simulations assuming synaptic transmission only, however, a larger difference between mutant numbers with and without complementation is observed (approximately 2-fold, Fig 4B), resulting from the frequent co-transmission of different virus strains, which occurs even at lower virus loads. Even more striking is the difference in the distribution of mutant numbers with and without complementation, under synaptic transmission (Fig 4B). The long

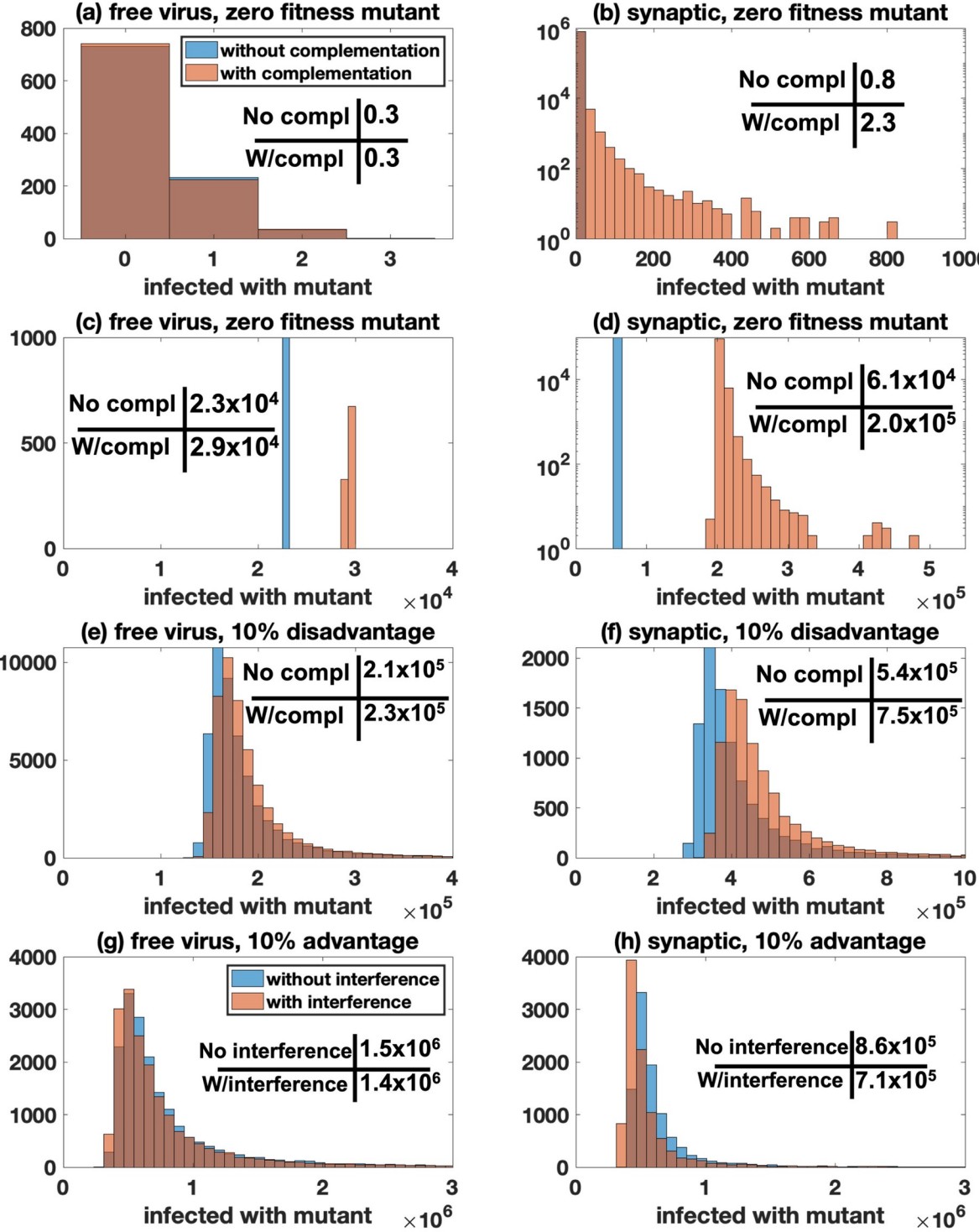

**Fig 4. Mutant evolution under different scenarios with 100% free virus transmission (left panels: $\beta = 3.6 \times 10^{-9}$, $\gamma = 0$, $N = 11$) or 100% synaptic transmission (right panels: $\beta = 0$, $\gamma = 3.6 \times 10^{-9}$, $S = 3$, $N = 25$).** Panels A and B record the number of cells infected with the mutant at $10^4$ infected cells, for all other panels it is $5 \times 10^8$ infected cells. For all panels, the blue bars represents simulations without complementation/interference and the red bars represents simulations with complementation/interference. The mean values are presented in each panel. For panels B-H, $p < 10^{-6}$ by the Kolmogorov-Smirnov test. A-D Zero fitness mutant ($F_{\text{mutant}} = 0$). E-F Disadvantageous mutant ($F_{\text{mutant}} = 0.81$). G-H Advantageous mutant ($F_{\text{mutant}} = 0.99$). For all simulations, we fix $\mathcal{M} = 50$, $F_{\text{wild-type}} = 0.9$ and the other parameters are as in Fig 1 ($\mu = 3 \times 10^{-5}$, $\lambda = 1.59 \times 10^7$, $a = 0.45$, and $d = 0.016$).

distribution tail in the presence of complementation is a result of early mutation events, which are extremely rare, but give rise to unusually high numbers of mutants at the threshold size. These events are similar to the so-called "jack-pot" event that have recently attracted attention in the context of mutant evolution in expanding cell populations [57, 58].

If the number of mutants is measured at higher virus loads, near peak, we find that complementation makes a modest difference if only free virus transmission is assumed (Fig 4C). This occurs because mutants that are generated at high virus loads will have a substantial chance to enter a cell that also contains a wild-type virus, leading to enhanced mutant spread at high virus loads. If we assume that the virus spreads only through synaptic transmission (Fig 4D), complementation makes a larger difference, but the effect of complementation is only slightly larger than that at low virus loads (Fig 4B). The reason is that the probability for wild-type and mutant viruses to be co-transmitted does not depend strongly on virus load.

We note that in the models with mutant generation, the effect of complementation on mutant numbers is much less pronounced than in simulations without mutation processes, even if the virus is assumed to only spread through virological synapses. The reason is that in the absence of mutational processes, the initially present mutant virus cannot spread without complementation, whereas it can do so in the presence of complementation. In the presence of mutational processes, however, even zero-fitness mutant numbers can rise over time without complementation, due to mutant production by wild-type viruses. Because the population size at peak virus load is large relative to the inverse of the mutation rate, mutant generation is a significant force that drives mutant numbers over time, limiting the difference that mutant replication in coinfected cells can make on the mutant population size.

Next, we assume that the mutant is no longer a zero-fitness type, but can be transmitted independently of the wild-type virus, although with a 10% fitness cost. In other words, if an infection event is attempted, it succeeds with a probability that is 10% smaller than that for the wild-type virus: $F_{\text{mutant}} = 0.9 F_{\text{wild-type}}$. If the mutant virus is in the same cell as the wild-type, however, this fitness cost is assumed to disappear and the mutant is neutral with respect to the wild-type virus. We focus on mutant numbers at high virus loads. We find that the number of mutants is only increased by a small amount, both if we assume that the virus spreads only by free virus transmission (Fig 4E) or only by synaptic transmission (Fig 4F); the difference is slightly larger for simulations that assume synaptic virus transmission, approximately 1.4 fold in Fig 4F).

The relatively small increase in mutant numbers brought about by complementation is surprising in the context of synaptic transmission. Intuitively, even though the disadvantageous mutant virus in Fig 4F can spread alone, the assumed 10% fitness cost, which is overcome by complementation, is still substantial. The reason for the limited impact of complementation is that in the presence of synaptic transmission, the actual fitness disadvantage of the mutant is reduced. The fitness cost is implemented by assuming that upon transfer to the new target cell, each virus has an increased probability to fail successful completion of infection. With synaptic transmission, it is assumed that there are $S$ infection attempts (in our simulation $S = 3$). This increases the likelihood that the cell will become infected (i.e. that at least one of the attempts is successful). Through this process, the effective fitness disadvantage of the mutant ends up being less than the 10% cost assumed per virus, which explains the modest effect of complementation on mutant numbers. The notion that the simultaneous transfer of multiple viruses per synapse reduces the effective relative fitness cost of a mutant has important implications that go beyond the scope of the current paper, and is explored in detail in a separate study. This analysis indicates that viral complementation might only make a substantial impact on the number of disadvantageous mutants if the disadvantage is very large. Therefore,

biologically, complementation might be most relevant to defective virus particles, and this effect is more pronounced under synaptic compared to free virus transmission.

Interference: Here we consider an advantageous mutant that loses fitness advantages in cells that contain both the mutant and the wild-type virus. This is implemented similarly to the simulations with disadvantageous mutants. To model the advantage, we assume that a mutant virus, upon transfer, succeeds in infecting the target cell with the probability that is 10% larger than that of the wild-type virus: $F_{\text{mutant}} = 1.1 F_{\text{wild-type}}$. As with complementation, Fig 4G and 4H shows that interference has a modest impact on the number of advantageous mutants at the size threshold (close to peak infection levels). Interference lowers the number of advantageous mutants to a slightly stronger degree if we assume synaptic (Fig 4H) rather than free virus transmission (Fig 4G), although the difference is relatively small in both cases, which is reminiscent of a similarly small effect of interference under synaptic transmission, observed in the absence of mutations, S13 Fig). The small effect for free virus transmission is explained by the absence of significant co-transmission of mutant and wild-type viruses, which limits the occurrence of the intracellular interactions among the two viral strains. For the simulations with synaptic transmission, the small effect is again explained by a reduction in the effective fitness difference between mutant and wild-type strains as a result of multiple, simultaneous infection events during synaptic transmission. Therefore, these results suggest that interference is unlikely to have a major impact on the dynamics of advantageous mutants, unless the advantage is very large, which would be biologically unrealistic (the simulations shown in Fig 4G and 4H already assume a 10% fitness advantage of the mutant).

## 4 Discussion

In this paper, we described a hybrid stochastic-deterministic algorithm to simulate viral evolutionary dynamics at large population sizes, including the occurrence of multiple infection of cells. The coevolution of relatively small populations (mutants and multiply infected cells) with larger populations (wild-type and singly infected cells) renders stochastic computer simulations computationally costly and not feasible when the virus population rises to higher levels. Ordinary differential equations can predict the average number of mutants over time, but can run into problems when describing the number of mutants at a given infected cell population size, the mutant number distributions, or the timing of mutant generation. The hybrid method described here, however, provides an accurate approximation of the true stochastic dynamics across all parameter ranges, at a fraction of the computational cost. This method therefore can serve as a practical tool to simulate complex viral evolutionary processes at large population sizes.

At the same time, however, the hybrid method can also run into computational limitations, depending the assumptions underlying the exact model formulation. While the hybrid method is capable of handling a large number of subpopulations, the number of "reactions" included in the stochastic part of the algorithm increases with (i) the number of different virus strains, (ii) the maximum multiplicity $N$, and (iii) the number of viruses transferred per synapse $S$. If these parameters are too large, the number of reactions for the Gillespie algorithm can become too high to be computationally feasible (even if only small populations are handled stochastically). In general, the number of strains per model is $2^k$ and the number of differential equations (subpopulations) per model is $\binom{N + 2^k}{2^k}$. If we model only free virus transmission, the number of infection events is the number of strains multiplied by the number of subpopulations eligible to be infected, but when synaptic transmission is included, there are many more infection events, which is correlated with the number of ways to partition $S$ into $2^k$ non-

negative integers that sum to 1, 2, . . ., $S$. When the number of reactions is on the order of $10^4$, each simulation becomes very computationally expensive, which happens, for example, if we consider triple mutants in the presence of synaptic transmission.

We used the hybrid stochastic-deterministic method to study how multiple infection and intracellular interactions among virus strains influence the evolutionary dynamics of mutants in the acute phase of HIV infection, during which the number of infected cells can rise to high levels, of the order of $10^8$ infected cells across the lymphoid tissues [14]. We showed that these processes can shape mutant evolution, but also found that this effect is restricted to select circumstances. On a basic level, the models confirmed the intuitive idea that multiple infection accelerates mutant evolution due to the larger number of mutation events during reverse transcription, when already infected cells become super-infected.

The model predictions about the ability of viral complementation to enhance the spread of disadvantageous mutants was more complex. According to the model, synaptic transmission is required to enhance disadvantageous mutant spread through complementation because it allows the repeated co-transmission of different virus strains; at the same time, however, this effect of complementation is only sizable if the selective disadvantage of the mutant is substantial, which most likely corresponds to a defective virus. The reason is that in the model studied here, synaptic transmission reduces the effective fitness difference between mutant and wild-type virus. This is because during a synaptic transmission event, multiple viruses are assumed to attempt infection of the target cells, thus increasing the chance that the cell will become infected with at least one of them. Even though we assumed a 10% lower probability of successful infection per mutant virus, in the context of our assumption that three viruses attempt infection per synapse, the overall chance that the cell becomes infected with a mutant is only 0.01% lower than the chance that it will become infected with a wild-type virus (the effective fitness difference). With a reduced effective fitness difference, complementation can only accelerate mutant growth by a modest amount.

Even if the effect of complementation is only pronounced for defective viruses, this still has strong biological significance. The maintenance of virus variants with zero or very low fitness during viral spread could be important for the evolvability of HIV in patients. The low fitness virus variants can potentially carry other mutations in their genomes, such as drug resistance or immune escape mutations. If these low fitness variants are repeatedly present in the same cell as wild-type viruses, recombination can transfer the mutation in question onto the wild-type genome, thus accelerating the rate of virus evolution. If the low fitness variants are not maintained, due to lack of complementation, however, this effect would not occur and could lead to a slower rate of virus evolution. Hence, maintenance of defective virus variants through complementation, and the consequent enhanced evolvability of the virus, could be one mechanism underlying the evolution of synaptic transmission in HIV infection. Recombination can be built into the models presented here to explore these dynamics in the future.

Another intracellular interaction that we considered was viral interference, where we track an advantageous mutant that loses fitness when together with a wild-type virus in an infected cell. As with complementation, for the fitness loss to be a driving event, the repeated co-transmission of wild-type and mutant virus is required through virological synapses. For the same reason as explained above, however, the multiple virus transfer events that occur during synaptic transmission reduce the fitness difference between the two virus strains, thus reducing the impact of interference on mutant numbers. To see a more significant effect would require a very substantial fitness advantage of the mutant, which is biologically unrealistic. According to our results, we therefore expect that viral interference is unlikely to significantly reduce the number of advantageous mutants.

According to the model studied here, viral complementation is not expected to play a significant role for mutant evolution in the absence of a transmission mechanism that involves the simultaneous transfer of multiple viruses from the infected cell to the target cell. It is important to remember, however, that the model presented here assumes well mixed virus and cell populations. If, in contrast, viruses spread in spatially structured cell populations with limited mixing, the spatial restriction could force the repeated co-transmission of different virus strains from one cell to another, even in the context of free virus transmission (simply because only a limited number of target cells are located in the immediate neighborhood of an infected cell). Therefore, spatial restriction during free virus transmission could have a similar effect as synaptic transmission during HIV infection. Indeed, computational modeling work has shown that similar to synaptic transmission, spatially restricted virus growth can lead to higher infection multiplicities, even at lower virus loads [59]. The correspondence between the properties of synaptic transmission in HIV infection and spatially restricted free virus spread remains to be established in more detail, and has relevance for a range of viral infections, importantly bacteriophage infections.

In addition, synaptic transmission of HIV might be characterized by spatial restrictions because an infected cell is most likely to form virological synapses with a neighboring target cell. We have previously investigated properties of spatially restricted synaptic virus transmission in a different context [31], and these models could be adapted to study the complementation and interference dynamics considered here. While synaptic transmission in HIV infection likely involves neighboring cells, however, CD4 T cells in the lymphoid tissue have been shown to move with a relatively fast rate [60], and the importance of spatial restrictions for the in vivo dynamics remains to be better understood.

## Supporting information

**S1 Text. Supplementary text and figures in support of the manuscript.**
(PDF)

**S1 Code. Hybrid program code written in C++.**
(CPP)

**S1 Fig. A comparison of the ODE prediction in Fig 1A in the main text to the corresponding system including explicit free virus equations; numerical values are also included in Table A in S1 Text.** The blue dots represent the deterministic prediction for the number of cells infected with the mutant strain when the total number of infected cells has reached $10^4$ (and are the same as the blue dots in Fig 1A in the main text), in the absence of the explicit free virus equations. The red stars represent the same quantity in the presence of the free virus equations. The parameters are $F_{\text{wild-type}} = 0.9$, $N = 3$, $\mu = 3 \times 10^{-5}$, $\lambda = 1.59 \times 10^7$, $\beta = 4 \times 10^{-9}$, $\gamma = 0$, $k = 2.25 \times 10^4$, $u = 500$, and $d = 0.016$. The infected cell death rate $a$ is adjusted to achieve the required $R_0$.
(PDF)

**S2 Fig. Histograms of the multiplicity of infection near peak infection.** The horizontal axis represents the average number of cells infected with the given number of viral copies, A for only free virus transmission, B for only synaptic transmission. S8 Fig also shows histograms for the average number of cells infected with the given number of viral copies for half free virus transmission and half synaptic transmission. The vertical axis is the average number of cells that are infected with different numbers of viral copies near peak infection. Infected with zero copies corresponds to the uninfected cells. Histograms were averaged over $10^2$ hybrid simulations with size threshold $\mathcal{M} = 50$. Simulations are stopped when the infected cell

population is near peak infection ($5 \times 10^8$ cells). Parameters are $\beta + \gamma = c = 3.6 \times 10^{-9}$, $\mu = 3 \times 10^{-5}$, $\lambda = 1.59 \times 10^7$, $a = 0.45$, and $d = 0.016$, and maximum multiplicity of infection $N$ is set to be large enough such that no cells reach this threshold.
(PNG)

**S3 Fig. Histograms of the number of cells infected with the neutral single and double mutant strain of the virus (double mutant model, $F = 1$ for all strains) when the infected cell population reaches $10^6$ cells for hybrid simulations with size threshold $\mathcal{M} = 50$ and $\mathcal{M} = 500$.** Simulations include only free virus transmission. For each size threshold $1.5 \times 10^5$ simulations were performed. A Number of cells infected with one of the single mutant strains. B Number of cells infected with the double mutant strain. The number of cells infected with the double mutant is not statistically different for $\mathcal{M} = 50$ and $\mathcal{M} = 500$ ($p > 0.1$ by Kolmogorov-Smirnov test). The parameters are $N = 3$, $\mu = 3 \times 10^{-5}$, $\lambda = 1.59 \times 10^7$, $\beta = 3.60 \times 10^{-9}$, $\gamma = 0$, $a = 0.45$, $d = 0.016$, and $R_0 = 8$.
(PNG)

**S4 Fig. Histograms of the number of cells infected with the neutral single mutant strain of the virus (single mutant model, $F_{\text{wild-type}} = F_{\text{mutant}} = 1$) when the infected cell population reaches $10^6$ cells for hybrid simulations with only free virus transmission.** Histograms for each size threshold represent $10^4$ simulations. The $p$-value from Kolmogorov-Smirnov test is shown for each comparison. A Size threshold $\mathcal{M} = 10$ and $\mathcal{M} = 500$. B Size threshold $\mathcal{M} = 50$ and $\mathcal{M} = 500$. C Size threshold $\mathcal{M} = 100$ and $\mathcal{M} = 500$. D Size threshold $\mathcal{M} = 200$ and $\mathcal{M} = 500$. Here $N = 1$ and the other parameters are $\mu = 3 \times 10^{-5}$, $\lambda = 1.59 \times 10^7$, $\beta = 3.60 \times 10^{-9}$, $\gamma = 0$, $d = 0.016$, and $R_0 = 1.5$.
(PNG)

**S5 Fig. Histograms of generation time of first single/double mutant virus in the double mutation model in the context of only free virus transmission (with all strains neutral).** Simulations in which infections are not established are discarded when calculating the averages. The deterministic prediction is denoted with the black vertical line and the hybrid average is denoted with the yellow vertical line. Histograms represent $10^5$ hybrid simulations with size threshold $\mathcal{M} = 50$. A Time until either single mutant generation. The deterministic prediction that the first single mutant virus (of both strains) will be generated is around 2.6 days, whereas in the stochastic case it is around 3 days. B Number of infected cells at first single mutant generation. The deterministic prediction is that the number of infected cells is around $3.5 \times 10^3$, whereas in the stochastic case it is around $1.4 \times 10^4$. C Time until double mutant generation. The deterministic prediction that the first double mutant virus will be generated is around 4.8 days, whereas in the stochastic case it is around 5.6 days. D Number of infected cells at first double mutant generation. The deterministic prediction is that the number of infected cells is around $3.5 \times 10^6$, whereas in the stochastic case it is around $4.3 \times 10^7$. The parameters are $N = 3$, $\mu = 3 \times 10^{-5}$, $\lambda = 1.59 \times 10^7$, $\beta = 3.60 \times 10^{-9}$, $\gamma = 0$, $a = 0.45$, $d = 0.016$, and $R_0 = 8$.
(PNG)

**S6 Fig. Comparing hybrid simulations with $\mathcal{M} = 50$ (red) with simulations in which the infected subpopulations are always treated stochastically (blue).** Simulations in which infections are not established are discarded when calculating the averages. Histograms represent $2 \times 10^4$ simulations. A The number of cells infected with the mutant at infected population size $10^4$. B The time at first mutant generation. The parameters are $F_{\text{wild-type}} = F_{\text{mutant}} = 1$, $N = 3$, $\mu = 3 \times 10^{-5}$, $\lambda = 1.59 \times 10^7$, $\beta = 3.60 \times 10^{-9}$, $\gamma = 0$, $a = 0.45$, $d = 0.016$, and $R_0 = 8$.
(PNG)

**S7 Fig. Deterministic time series evolution of an infection with a neutral mutant strain with only free virus (dashed lines, $\gamma = 0$) or only synaptic transmission (solid lines, $\beta = 0$).** Parameters are $N = 25$, $\beta + \gamma = c = 3.6 \times 10^{-9}$, $\mu = 3 \times 10^{-5}$, $\lambda = 1.59 \times 10^{7}$, $a = 0.45$, and $d = 0.016$. The multiplicity of infection is shown with the red lines, the number of cells infected with only the wild-type virus are shown with the yellow lines, the number of cells infected with only the mutant are shown with the purple lines, and the number of cells coinfected with both the wild-type and mutant are shown with the green lines.
(PNG)

**S8 Fig. Histograms of the multiplicity of infection near peak infection.** A Histograms for the average number of cells infected with the given number of viral copies for only free virus transmission. B Histograms for the average number of cells infected with the given number of viral copies for half free virus transmission and half synaptic transmission. The horizontal axis is the number of virus copies and the vertical axis is the average number of cells that are infected with that number of viral copies near peak infection. Infected with zero copies corresponds to the uninfected cells. Histograms were averaged over $5 \times 10^{2}$ hybrid simulations with size threshold $\mathcal{M} = 50$. Simulations are stopped when the infected cell population is close to peak infection ($6 \times 10^{8}$ cells). Parameters are $\beta + \gamma = c = 3.6 \times 10^{-9}$, $\mu = 3 \times 10^{-5}$, $\lambda = 1.59 \times 10^{7}$, $a = 0.45$, and $d = 0.016$, and maximum multiplicity of infection $N$ is set to be large enough such that no cells reach this threshold.
(PNG)

**S9 Fig. Neutral mutant evolution in the absence of synaptic transmission, comparing simulations with single infection only ($N = 1$, blue) and in the presence of multiple infection ($N = 5$, red).** For both panels, the Kolmogorov-Smirnov test between the two distributions suggests that they are not statistically different. A Number of cells infected with one of the single mutant strains. B Number of cells infected with the double mutant strain. Distributions represent $4.5 \times 10^{4}$ hybrid simulations with size threshold $\mathcal{M} = 50$. Simulations are stopped at a low viral load ($10^{5}$ cells). The other parameters are as in main text Fig 2 ($F_{\text{wild-type}} = 1$, $F_{\text{mutant}} = 1$, $\mu = 3 \times 10^{-5}$, $\lambda = 1.59 \times 10^{7}$, $\beta = 3.60 \times 10^{-9}$, $\gamma = 0$, $a = 0.45$, and $d = 0.016$).
(PNG)

**S10 Fig. Non-neutral (disadvantageous and advantageous) mutant evolution in the absence of synaptic transmission, comparing simulations with single infection only ($N = 1$, blue) and in the presence of multiple infection ($N = 11$, red).** The mean values are shown by black the vertical lines (blue for single infection only and red for multiple infection). For both panels, the Kolmogorov-Smirnov test between the two distributions gives a $p$-value less than $10^{-6}$. A Disadvantageous mutant; here $F_{\text{mutant}} = 0.81$. The average under single infection only is approximately $1.6 \times 10^{5}$ and for multiple infection is approximately $3.2 \times 10^{5}$. B Advantageous mutant with interference; here $F_{\text{mutant}} = 0.99$. The average for single infection is approximately $1.2 \times 10^{6}$ and for multiple infection is approximately $2.3 \times 10^{6}$. Histograms represent $2 \times 10^{4}$ hybrid simulations with size threshold $\mathcal{M} = 50$. Simulations are stopped when the infected cell population is close to peak infection ($6 \times 10^{8}$ cells). The other parameters are $F_{\text{wild-type}} = 0.9$, $\mu = 3 \times 10^{-5}$, $\lambda = 1.59 \times 10^{7}$, $\beta = 4 \times 10^{-9}$, $\gamma = 0$, $a = 0.45$, and $d = 0.016$.
(PNG)

**S11 Fig. Presence of the triple mutant strain in the absence of synaptic transmission, comparing simulations with single infection only ($N = 1$, blue) and in the presence of multiple infection ($N = 3$, red).** The probability to have at least one cell infected by a triple-mutant is 3.3% under multiple infection, which is about 1.7 times higher than that under single infection

(1.9%). This result is significant with $p = 2.5 \times 10^{-3}$ by the Z-test, with $2.7 \times 10^4$ runs under single infection and $1.9 \times 10^3$ runs under multiple infection. Histograms represent $10^3$ hybrid simulations with size threshold $\mathcal{M} = 50$. Simulations are stopped when the infected cell population is close to peak infection ($6 \times 10^8$ cells). All strains are neutral ($F = 1$) and all other parameters are $\mu = 3 \times 10^{-5}$, $\lambda = 1.59 \times 10^7$, $\beta = 3.60 \times 10^{-9}$, $\gamma = 0$, $a = 0.45$, and $d = 0.016$.
(PNG)

**S12 Fig. Neutral mutant evolution in the presence of synaptic transmission, comparing simulations with different combinations of free virus and synaptic transmission (single mutation only).** A Number of cells infected with the mutant strain under different transmission strategies. The horizontal axis is the percent contribution of free virus transmission. Standard error bars are shown in the main figure, and standard deviation bars are shown in the inset. B Histograms for the number of cells infected with a single mutant for the different strategies, representing $10^4$ hybrid simulations with size threshold $\mathcal{M} = 50$. Simulations are stopped when the infected cell population is close to peak infection ($6 \times 10^8$ cells) and simulations where no infection is established are thrown out. Here $F_{\text{wild-type}} = 1$, $F_{\text{mutant}} = 1$, $S = 3$, $N = 25$, $\beta + \gamma = c = 3.6 \times 10^{-9}$, and the other parameters are $\mu = 3 \times 10^{-5}$, $\lambda = 1.59 \times 10^7$, $a = 0.45$, and $d = 0.016$.
(PNG)

**S13 Fig. 10% advantageous mutant ($F_{\text{wild-type}} = 0.9$, $F_{\text{mutant}} = 0.99$), comparing the effect of interference and free virus versus synaptic transmission near peak infection, in the absence of mutations ($\mu = 0$).** All simulations start with a single infected cell coinfected with a single copy of both the wild-type and mutant. Panels A and B represent only free virus transmission ($\beta = 3.60 \times 10^{-9}$, $\gamma = 0$, $N = 11$), whereas panels C and D represent only synaptic transmission ($\beta = 0$, $\gamma = 3.60 \times 10^{-9}$, $N = 25$). The Kolmogorov-Smirnov test between panels A and B and panels C and D gives a $p$-value less than $10^{-6}$. A Only free virus transmission without interference. The average number of cells infected with the mutant is $3.8 \times 10^8$. B Only free virus transmission with interference. The average number of cells infected with the mutant is $3.6 \times 10^8$. C Only synaptic transmission without interference. The average number of cells infected with the mutant is $3.4 \times 10^8$. D Only synaptic transmission with interference. The average number of cells infected with the mutant is $3.1 \times 10^8$. Histograms represent $6 \times 10^3$ hybrid simulations with size threshold $\mathcal{M} = 50$. Simulations in which infections are not established (or in the rare case a simulation does not reach the infected size threshold) are discarded; simulations are stopped when the infected cell population is close to peak infection ($5 \times 10^8$ cells). The other parameters are $\lambda = 1.59 \times 10^7$, $a = 0.45$, and $d = 0.016$.
(PNG)

## Author Contributions

**Conceptualization:** Jesse Kreger, Natalia L. Komarova, Dominik Wodarz.

**Formal analysis:** Jesse Kreger, Natalia L. Komarova, Dominik Wodarz.

**Funding acquisition:** Natalia L. Komarova, Dominik Wodarz.

**Investigation:** Jesse Kreger, Natalia L. Komarova, Dominik Wodarz.

**Methodology:** Jesse Kreger, Natalia L. Komarova, Dominik Wodarz.

**Software:** Jesse Kreger.

**Writing – original draft:** Jesse Kreger, Natalia L. Komarova, Dominik Wodarz.

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
