## [Decision Letter · Decision Letter 0]

11 May 2021

Dear Mr. Kreger,

Thank you very much for submitting your manuscript "A hybrid stochastic-deterministic approach to explore multiple infection and evolution in HIV" for consideration at PLOS Computational Biology.

As with all papers reviewed by the journal, your manuscript was reviewed by members of the editorial board and by several independent reviewers. In light of the reviews (below this email), we would like to invite the resubmission of a significantly-revised version that takes into account the reviewers' comments.

There are important comments from the referees on the novelty of the methodology, on the biological assumptions, and on the emphasis on the methodology vs. biological results. There are also concerns about the structure and clarity of the manuscript.  A more clear exposition of both, and a further emphasis on the contributions on HIV, seem to be required. 

We cannot make any decision about publication until we have seen the revised manuscript and your response to the reviewers' comments. Your revised manuscript is also likely to be sent to reviewers for further evaluation.

Sincerely,

Mercedes Pascual

Associate Editor

PLOS Computational Biology

Nina Fefferman

Deputy Editor

PLOS Computational Biology

Reviewer's Responses to Questions

**Comments to the Authors:**

Reviewer #1: This paper describes a hybrid stochastic/deterministic approach applied to HIV. The underlying model studied takes into account synaptic cell-to-cell transmission (for multiple strain infections), and neutral, and advantageous/disadvantageous mutations. ODE models are widely adopted to study viral dynamics, despite their downsides when it comes to rare mutational events. The approach described herein therefore has the potential for impact, given that it addresses such events during acute infection, as described in this paper.

1) This paper would benefit significantly from reorganization. If the emphasis is on the computational method, then parts of the SI should be moved to the main paper. I believe this would help the reader better grasp the approach. I would suggest at a minimum that the first section of the SI (full description of the mathematical model) be moved into the main paper, to give more context to the later sections. As written, the main takeaways and the focus of the paper aren't clearly distinguishable.

2) The authors should touch upon why cells non-dimensionalized in the model. Would this be necessary for all implementations of the hybrid approach? Later in figure legends, cells are expressed in cells. If cell units are not not non-dimensionalized, than the units in Table 1 aren't correct.

3) A more detailed discussion on other hybrid approaches should be undertaken in the introduction. These include, but aren't limited to, tau-leaping and fast-slow systems.

4) Why is free virus not explicitly included in the model (Eqs. 1-5)? As written, the rate of free virus infection is proportional to beta and interactions between uninfected and all infected cells, why not include an equation for virions?

5) Figure 1 and similar figures: the dots are quite large and the inner panel is doubling up information. Since multiple runs of the stochastic algorithm are needed, it would be better to always include mean + SD (or some other metric of dispersion). I would suggest remaking these large dots with another smaller marker and tails.

6) Figure 2: What do the royal blue and bright red single lines represent?

Reviewer #2: Please see the attache pdf file of the review report.

Reviewer #3: Please see attached pdf.

**Have the authors made all data and (if applicable) computational code underlying the findings in their manuscript fully available?**

Reviewer #1: Yes

Reviewer #2: Yes

Reviewer #3: None

PLOS authors have the option to publish the peer review history of their article (what does this mean?). If published, this will include your full peer review and any attached files.

Reviewer #1: No

Reviewer #2: No

Reviewer #3: No
---

## [Decision Letter · Decision Letter 1]

25 Oct 2021

Dear Mr. Kreger,

Thank you very much for submitting your manuscript "A hybrid stochastic-deterministic approach to explore multiple infection and evolution in HIV" for consideration at PLOS Computational Biology. Your manuscript was reviewed by members of the editorial board and by the same independent reviewers who commented on your original submission. The reviewers indicated that their concerns have now been largely addressed.  On this basis, we are  ready to accept this manuscript for publication once you consider and clarify two pending issues they have raised, on  the description of the algorithm and Piecewise Deterministic Markov Processes and on the units on the table.  It would be worthwhile to acknowledge the literature on efficient algorithms for PDMP and to discuss your algorithm in that context 

[1] A letter containing your responses to the above two comments, and a brief description of the changes you have made in the manuscript. Please note while forming your response, if your article is accepted, you may have the opportunity to make the peer review history publicly available. The record will include editor decision letters (with reviews) and your responses to reviewer comments. If eligible, we will contact you to opt in or out

Sincerely,

Mercedes Pascual

Associate Editor

PLOS Computational Biology

Nina Fefferman

Deputy Editor

PLOS Computational Biology

[LINK]

Reviewer's Responses to Questions

**Comments to the Authors:**

Reviewer #1: The authors have made great improvements to the flow in their revisions, and have responded to my previous questions. I am not sure why the variables with * in Table 1 do not simply have units in the units column instead of the note in the caption, but I assume the authors have a specific reason for this choice.

Reviewer #2: Authors have addressed all my comments. I thank them for the updated work. I do not have any further comments and recommend for publication in PLOS Computational Biology.

Reviewer #3: Please see the comment in the attached PDF.

**Have the authors made all data and (if applicable) computational code underlying the findings in their manuscript fully available?**

Reviewer #1: Yes

Reviewer #2: Yes

Reviewer #3: Yes

PLOS authors have the option to publish the peer review history of their article (what does this mean?). If published, this will include your full peer review and any attached files.

Reviewer #1: No

Reviewer #2: No

Reviewer #3: No

Figure Files:

Data Requirements:

Reproducibility:

References:

---

## [Editor Report · Decision Letter 2]

2 Dec 2021

Dear Mr. Kreger,

We are pleased to inform you that your manuscript 'A hybrid stochastic-deterministic approach to explore multiple infection and evolution in HIV' has been provisionally accepted for publication in PLOS Computational Biology.

Best regards,

Mercedes Pascual

Associate Editor

PLOS Computational Biology

Nina Fefferman

Deputy Editor

PLOS Computational Biology

---

## [Editor Report · Acceptance letter]

16 Dec 2021

PCOMPBIOL-D-21-00304R2 

A hybrid stochastic-deterministic approach to explore multiple infection and evolution in HIV

Dear Dr Kreger,

I am pleased to inform you that your manuscript has been formally accepted for publication in PLOS Computational Biology. Your manuscript is now with our production department and you will be notified of the publication date in due course.

With kind regards,

Agnes Pap
